# TILP: Differentiable Learning of Temporal Logical Rules on Knowledge Graphs

**Siheng Xiong, Yuan Yang, Faramarz Fekri & James Clayton Kerce**
Georgia Institute of Technology
Atlanta, GA 30332, USA
`{sxiong45,yyang754}@gatech.edu, faramarz.fekri@ece.gatech.edu,`
`clayton.kerce@gtri.gatech.edu`

## Abstract

Compared with static knowledge graphs, temporal knowledge graphs (tKG), which can capture the evolution and change of information over time, are more realistic and general. However, due to the complexity that the notion of time introduces to the learning of the rules, an accurate graph reasoning, e.g., predicting new links between entities, is still a difficult problem. In this paper, we propose TILP, a differentiable framework for temporal logical rules learning. By designing a constrained random walk mechanism and the introduction of temporal operators, we ensure the efficiency of our model. We present temporal features modeling in tKG, e.g., recurrence, temporal order, interval between pair of relations, and duration, and incorporate it into our learning process. We compare TILP with state-of-the-art methods on two benchmark datasets. We show that our proposed framework can improve upon the performance of baseline methods while providing interpretable results. In particular, we consider various scenarios in which training samples are limited, data is biased, and the time range between training and inference are different. In all these cases, TILP works much better than the state-of-the-art methods.

## 1 Introduction

Knowledge graphs (KGs) contain facts $(e_s, r, e_o)$ representing relation $r$ between subject entity $e_s$ and object entity $e_o$, e.g., (David Beckham, plays for, Real Madrid). In real world, many relations are time-dependent, e.g., a player joining a team for a season, a politician holding a position for a certain period of time, and two persons' marriage lasting for decades. To represent the evolution and change of information, temporal knowledge graphs (tKGs) have been introduced. An interval $I$, indicating the valid period of the fact, is utilized by tKGs to extend the triples $(e_s, r, e_o)$ into quadruples $(e_s, r, e_o, I)$, e.g., (David Beckham, plays for, Real Madrid, [2003, 2007]).

Automatically reasoning over KGs such as link predication, i.e., inferring missing facts using existing facts, is a common task for real-world applications. However, the introduction of temporal information makes this task more difficult. The important dynamic interactions between entities can not be captured by learning methods developed for static KGs. Recently, a few embedding-based frameworks have been proposed to address the above limitation, e.g., HyTE (Dasgupta et al. (2018)), TNTComplEx (Lacroix et al. (2020)), and DE-SimplE (Goel et al. (2019)). The common principle adopted by these models is to create time-dependent embeddings for entities and relations.

Alternatively, first-order inductive logical reasoning methods have some desirable features relative to embedding methods when applied to KGs, as they provide interpretable and robust inference results. Since the resulting logical rules contain temporal information in tKGs, we call them temporal logical rules. Some recent works, e.g., StreamLearner (Omran et al. (2019)), and TLogic (Liu et al. (2021)), have introduced a framework for temporal KG reasoning. However, there are still several unaddressed issues. First, these statistical methods count from graph the number of paths that support a given rule as its confidence estimation. As such, this independent rule learning ignores the interactions between different rules from the same positive example. For instance, given certain rules, the confidence of some rules might be enhanced, while that of others can be diminished. Sec-

ond, these methods cannot deal with the similarity between different rules. Given a reliable rule, it is reasonable to believe that the confidence of another similar rule, e.g., with the same predicates but slightly different temporal patterns, is also high. However, its estimated confidence with these methods can be quite low if it is infrequent in the dataset. Finally, the performance of these timestamp-based methods on interval-based tKGs is not demonstrated. It should be noted that the temporal relations between intervals are more complex than those of timestamps. All these problems are solved by our neural-network-based framework.

In this paper, we propose TILP, a differentiable inductive learning framework. TILP benefits from a novel mechanism of constrained random walk and an extended module for temporal features modeling. We achieve comparable performance to the state-of-the-art methods, while providing logical explanations for the inference results. More specifically, our main contributions are summarized as follows:

- TILP, a novel differentiable and temporal inductive logic framework, is introduced based on constrained random walks on temporal knowledge graphs and temporal features modeling. It is the first differentiable approach that can learn temporal logical rules from tKGs without restrictions.

- Experiments on two benchmark datasets, i.e., WIKIDATA12k and YAGO11k, are conducted, where our framework shows comparable or improved performance relative to the state-of-the-art methods. For test queries, our framework has the advantage that it provides both the ranked list of candidates and explanations for the prediction.

- The superiority of our method compared to existing methods is demonstrated in several scenarios such as when training samples are limited, data is biased, and time range of training and testing are different.

## 2 RELATED WORKS

**Embedding-based methods.** Recently, embedding-based methods for tKGs started emerging for a more accurate link prediction. The common principle of these methods is to create time-dependent embeddings for entities and relations, e.g., HyTE (Dasgupta et al. (2018)), TA-ComplEx (García-Durán et al. (2018)), TNTComplEx (Lacroix et al. (2020)), and DE-SimplE (Goel et al. (2019)). These embeddings are plugged into standard scoring functions in most cases. Further, other works, e.g, TAE-ILP (Jiang et al. (2016)), and TimePlex (Jain et al. (2020)), have investigated the explicit temporal feature modeling, and merged it into the embedding algorithms. The main weakness of embedding-based methods is lack of interpretability, as well as their failure when previously unobserved entities, relations, or timestamps present during inference.

**Logical-rule-based methods.** Logical-rule-based methods for link prediction on tKGs is mainly based on random walks. Although these works show the ability of learning temporal rules, they perform random walks in a very restricted manner, which impairs the quality of learned rules. For example, Dynnode2vec (Mahdavi et al. (2018)) and Change2vec (Bian et al. (2019)) both process tKGs as a set of graph snapshots at different times where random walks are performed separately. DynNetEmbedd (Nguyen et al. (2018)) requires the edges in walks to be forward in time. StreamLearner (Omran et al. (2019)) first extracts rules from the static random walk, and then extend them separately into time domain. The consequence is that all body atoms in the extended rules have the same timestamp. TLogic (Liu et al. (2021)) is the most recent work which extracts temporal logical rules from the defined temporal random walks. The temporal constraints for temporal random walks are built on timestamps instead of intervals, and are fixed during the learning. This inflexibility impairs its ability in temporal constraints learning. Furthermore, both StreamLearner and TLogic, which can truly learn temporal logical rules, are statistical methods which estimates the rule confidence by counting the number of rule groundings and body groundings.

**Differentiable rule learning.** Several works utilize neural network architectures for rule learning, e.g., Neural-LP (Yang et al. (2017)), NTP (Rocktäschel & Riedel (2017)), DeepProblog (Manhaeve et al. (2018)), $\partial$ILP (Evans & Grefenstette (2018)), RuLES (Ho et al. (2018)), IterE (Zhang et al. (2019)), dNL-ILP (Payani & Fekri (2019)) and NLProlog (Weber et al. (2019)). These works mainly focus on static KGs, lacking the ability of capturing temporal patterns. Converting from static KGs to temporal KGs is not a trivial extension. For example, Neural-LP reduces the rule learning problem to

matrix multiplication. By defining the operators and neural control system, this framework realizes logical rule learning in an end-to-end fashion. However, the extra temporal constraints in logical rules break down the Markovian property of random walks which serves as the foundation of the past frameworks.

## 3 PRELIMINARIES

**Temporal knowledge graph.** A temporal knowledge graph (tKG) $\mathcal{G}$ is a collection of facts represented by a quadruple $(e_s, r, e_o, I)$. This fact, also called edge or link, implies a relation $r$ from the subject entity $e_s$ to the object entity $e_o$ during interval $I$. We define an interval $I$ with its start time $t_s$ and end time $t_e$, i.e., $I = [t_s, t_e]$. To allow bidirectional random walks, we imagine the existence of inverse edges, i.e., $(e_o, r^{-1}, e_s, I)$. The set of entities, relations, timestamps and intervals are denoted by $\mathcal{E}, \mathcal{R}, \mathcal{T}$ and $\mathcal{I}$, respectively.

**Link prediction.** This task is to predict missing links with observed facts from the same tKG. To be specific, given a query $(e_s, r, ?, I)$ and the observed facts from the same tKG $\mathcal{G}$, a ranked list of candidates for the missing object is required. For subject prediction, the query is formulated as $\left(e_o, r^{-1}, ?, I\right)$. Compared with static link predication, the temporal link prediction is much harder: Even given the same subject (or object) and relation, the correct answer can change with different query intervals.

**Temporal relation.** Temporal relation (TR) between two timestamps $t$ and $t'$ can take the form $before$, $equal$, or $after$, which are denoted by $t < t'$, $t = t'$, and $t > t'$, respectively. For TR between two intervals $I$ and $I'$, there are 13 different relations in total given by Allen's interval algebra (Allen (1983)). For example, $I := [t_s, t_e]$ is $before$ $I' := [t'_s, t'_e]$ iff $t_e < t'_s$. However, the resulting temporal logical rules would become too specific by directly using all these 13 types. Thus, we group them into 3 classes: $TR \in \{before, touching, after\}$, where $TR$ denotes the possible TR between two intervals, $before$ is defined as previously mentioned, $after$ is the converse of $before$, and $touching$ is the group of all the other 11 types. In other words, touching is used when two intervals have overlap with each other. It should be noted that timestamp can be considered as a special kind of interval with equal start time and end time. Thus, our definition of temporal relation $TR$ can be also used to describe TR between timestamps.

**Temporal logical rule.** A temporal logical $Rule$ of length $l \in \mathbb{N}$ is defined as

$$P_{l+1}(E_1, E_{l+1}, I_{l+1}) \leftarrow \wedge_{i=1}^{l} P_i(E_i, E_{i+1}, I_i) \wedge_{j=1}^{l} \left(\wedge_{k=j+1}^{l+1} TR_{j,k}(I_j, I_k)\right) \tag{1}$$

where $E_i \in \mathcal{E}$ denotes variables of entities, $I_i \in \mathcal{I}$ denotes variables of intervals, $P_i \in \mathcal{R}$ denotes predicates, which are grounded relations in logical rules, and $TR_{j,k} \in \{before, touching, after\}$ denotes grounded temporal relations between intervals.

The left arrow in $Rule$ is called "entails", i.e., the rule body on the right entails the rule head on the left. The rule head contains a head predicate $P_{l+1}$, also called the target predicate. For the link prediction task, target predicates are given. Thus, we use $P_h$ to denote it in the following sections. $I_{l+1}$ is called query interval. Similarly, $P_i$ and $I_i$ for $i \in [1, l]$ are called body predicates and body intervals, respectively. The rule is also called "chain-like" because the rule body corresponds to a walk from $E_1$ to $E_{l+1}$.

A rule is grounded by substituting the variables $E$ and $I$ with constants. For example, a grounding of the following temporal logical rule

$$\text{ReceiveAward}(E_1, E_2, I_2) \leftarrow \text{NominatedFor}(E_1, E_2, I_1) \wedge \text{touching}(I_1, I_2)$$

is given by the edges (Alice Bradley Sheldon, receive award, Nebula Award for Best Novelette, [1977, 1977]) and (Alice Bradley Sheldon, nominated for, Nebula Award for Best Novelette, [1977, 1977]) in WIKIDATA12k dataset. Since logical rules can be violated, rule confidence, the probability that a rule is correct, needs to be estimated.

## 4 CONSTRAINED RANDOM WALK

**Path constraint.** Temporal logical rules can be considered as constraints for random walks on tKG. Generally speaking, these constraints can be divided into two classes: Markovian and non-

Markovian. With Markovian constraints, the calculation of next state probability is only related to current state probability, i.e., a random walk is performed without the consideration of previous visited edges. Otherwise, we need to record the previous visited edges to ensure that non-Markovian constraints are satisfied.

For the temporal logical rule given in (1), Markovian constraints include body predicates, i.e., $P_i$ for $i \in [1, l]$, and TRs between query interval and every body interval, i.e., $TR_{i,l+1}$ for $i \in [1, l]$. Further, the non-Markovian constraints include pairwise TRs between body intervals, i.e., $TR_{j,k}$ for $j \in [1, l-1]$ and $k \in [j+1, l]$. Filtering operators $f$ for these constraints are defined as

$$f_P\left((e_s, r, e_o, I)\right) = \begin{cases} 1 & \text{if } r = P, \\ 0 & \text{otherwise.} \end{cases} \tag{2}$$

$$f_{TR}\left((e_s, r, e_o, I), (e_s', r', e_o', I')\right) = \begin{cases} 1 & \text{if } g(I, I') = TR, \\ 0 & \text{otherwise.} \end{cases} \tag{3}$$

$$g\left(I, I'\right) = \begin{cases} \text{before} & \text{if } t_e < t_s', \\ \text{after} & \text{if } t_s > t_e', \\ \text{touching} & \text{otherwise.} \end{cases} \tag{4}$$

where $f_P$ and $f_{TR}$ denote filtering operators for predicate $P$ and temporal relation $TR$, respectively, and $g$ is the TR evaluation function with $I := [t_s, t_e]$ and $I' := [t_s', t_e']$.

**Constrained random walk.** Since a successful random walk should satisfy both classes of constraints, we first perform random walk under Markovian ones, and then filter out the results according to non-Markovian ones. To ensure the efficiency of our framework, we use matrix operators built from the filtering operators. Given a query $(e_s, r, ?, I)$, for every pair of entities $e_x, e_y \in \mathcal{E}$, the operator $M_{i,CM_i} \in \{0, 1\}^{|\mathcal{E}| \times |\mathcal{E}|}$ related to step $i$ under corresponding Markovian constraints $CM_i = \{P_i, TR_{i,l+1}\}$ is defined as:

$$\begin{aligned} (M_{i,CM_i})_{x,y} &= \max_{F \in \mathcal{F}_{y,x}} f_{CM_i}(F) \\ &= \max_{F \in \mathcal{F}_{y,x}} f_{P_i}(F) \, f_{TR_{i,l+1}}(F, (e_s, r, ?, I)) \end{aligned} \tag{5}$$

where $(M_{i,CM_i})_{x,y}$ denotes the $(x, y)$ entry of $M_{i,CM_i}$, $F$ denotes a single fact, and $\mathcal{F}_{y,x}$ denotes the set of facts from $e_y$ to $e_x$. The essence of the operator is the adjacency matrix under Markovian constraints, and we set the entry maximum to 1. With these operators, we can actually find all the paths between any pair of entities that satisfy these Markovian constraints. Suppose we start from entity $e_s$. After $l$ steps of random walk under corresponding Markovian constraints, the process is written as

$$\mathbf{v}_{i+1} = M_{i,CM_i} \mathbf{v}_i \qquad \text{for } 1 \le i \le l \tag{6}$$

where $\mathbf{v}_i \in \mathbb{N}^{|\mathcal{E}|}$ is the indicator vector, e.g., for $\mathbf{v}_1$ only the entry related to $e_s$ is set to 1, other entries being 0. When arriving at some entities, the corresponding entries would be greater than 0. With these indicator vectors, we can obtain every single constrained random walk $W_{CM}^{(n)}$ for $n \in \mathbb{N}$ given by $((e_1^{(n)}, r_1^{(n)}, e_2^{(n)}, I_1^{(n)}), \ldots, (e_l^{(n)}, r_l^{(n)}, e_{l+1}^{(n)}, I_l^{(n)}))$.

For non-Markovian constraints $CN_{j,k} = \{TR_{j,k}\}$, we apply corresponding filtering functions to these walks as:

$$\begin{aligned} f_{CN}(W_{CM}^{(n)}) &= \prod_{j=1}^{l-1} \prod_{k=j+1}^{l} f_{CN_{j,k}}((e_j^{(n)}, r_j^{(n)}, e_{j+1}^{(n)}, I_j^{(n)}), (e_k^{(n)}, r_k^{(n)}, e_{k+1}^{(n)}, I_k^{(n)})) \\ &= \prod_{j=1}^{l-1} \prod_{k=j+1}^{l} f_{TR_{j,k}}((e_j^{(n)}, r_j^{(n)}, e_{j+1}^{(n)}, I_j^{(n)}), (e_k^{(n)}, r_k^{(n)}, e_{k+1}^{(n)}, I_k^{(n)})) \end{aligned} \tag{7}$$

The set of filtered walks $S_{W_C} := \{W_{CM}^{(n)} : f_{CN}(W_{CM}^{(n)}) = 1\}$ is the final result of our algorithm. In this process, we also remove walks that involve repeated edges. In fact, by introducing new filtering functions, this framework can have more possibilities. For example, using the numerical comparison filtering functions (Wang et al. (2019)), our method is able to learn logical rules with numerical features such as a person's age, height and weight.

## 5 LEARNING TEMPORAL LOGICAL RULES

The framework mainly consists of two stages: rule learning followed by rule application. In the learning stage, given a set of positive examples for the target predicates, we are going to find all the paths from the subject entity to the object entity in the tKG. Then we extract temporal logical rules from these paths, and estimate their confidence. The parameterised distributions of different temporal features introduced in this sections are also measured at this stage.

In the application stage, given a query and a set of temporal logical rules related to the target predicate, we are going to find all the random walks that satisfy these rules. By calculating the arriving rate of each rule and aggregating all the rules, we can obtain the temporal logical rule score for all candidates. In addition, we use all the evidences related to these candidates to evaluate their temporal feature scores which, together with temporal logical rule scores, form the final scores.

**Rule confidence.** For the extracted temporal logical rules, we need to estimate their confidence for the rule application. Instead of estimating confidence for every single rule, we create attention vectors for every type of constraints, and re-use them in different rules. Sharing confidence of constraints creates a joint and robust learning process, and largely reduces model parameters.

In our temporal logical rules, there are two types of constraints, predicates and temporal relations. For every target predicate, we create a set of attention vectors to denote the confidence of using a certain constraint. Obviously, there are many factors that can affect these attention vectors, e.g, the target predicate, the query interval, the rule length, entity properties, and so on. To make this problem tractable, some simplifications are made here. We suppose that the attention vectors of predicates and TRs are dependent on the target predicate and the rule length. Furthermore, to deal with varying lengths of rules, an attention vector of the rule length is also required. Inspired by Neural-LP (Yang et al. (2017)), we design a set of mapping functions based on RNN.

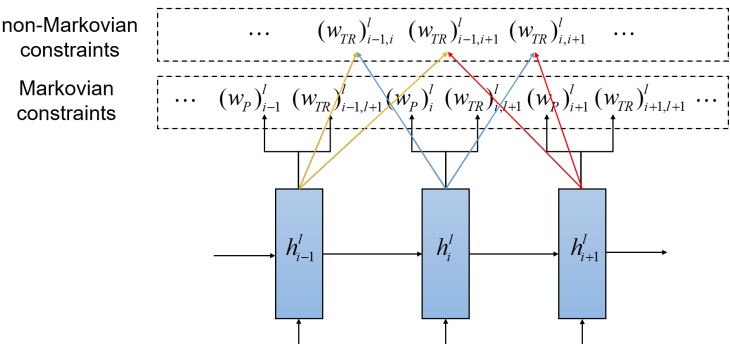

Figure 1: The RNN-based system for solving the attention vectors

Specifically, as shown in Fig.1, let $w_{Len} \in \mathbb{R}^L$ be the attention vector of rule length where $L$ is the maximum. For a certain length $l \in [1, L]$, at each step $i \in [1, l]$ we calculate the attention vector of predicate $(w_P)_i^l \in \mathbb{R}^{|\mathcal{R}|}$ and its TR between query interval $(w_{TR})_{i,l+1}^l \in \mathbb{R}^{|TR|}$. We also calculate the attention vector of pairwise TR between body intervals $(w_{TR})_{j,k}^l \in \mathbb{R}^{|TR|}$ for $j \in [1, l-1]$ and $k \in [j+1, l]$ using corresponding states.

$$\mathbf{h}_i^l = \text{update}\left(\mathbf{h}_{i-1}^l, f_{embdd}^{\mathrm{T}}(\mathbf{X}^l, P_h)\right) \tag{8}$$

$$(w_P)_i^l = \text{softmax}\left(W_P \mathbf{h}_i^l + b_P\right) \tag{9}$$

$$(w_{TR})_{i,l+1}^l = \text{softmax}\left(W_{TR} \mathbf{h}_i^l + b_{TR}\right) \tag{10}$$

$$(w_{TR})_{j,k}^l = \text{softmax}\left(W_{TR}'[\mathbf{h}_j^l; \mathbf{h}_k^l] + b_{TR}'\right) \tag{11}$$

$$w_{Len} = \text{softmax}\left(W_{Len} f_{embdd}^{\mathrm{T}}(\mathbf{X}^0, P_h) + b_{Len}\right) \tag{12}$$

where $\mathbf{h}_i^l \in \mathbb{R}^d$ denotes the $i$th-step state in the $l$-length rule with feature dimension $d$, $\mathbf{h}_0^l$ is set as $\mathbf{0}$, $\mathbf{X}^l \in \mathbb{R}^{|\mathcal{R}| \times d}$ denotes the embedding matrix for target predicate $P_h$, $f_{embdd}(\mathbf{X}, p) := u_p^T \mathbf{X}$ is an embedding lookup function in which $u_p$ is a one-hot indicator vector for input $p$, i.e.,

$f_{embdd}^{\mathrm{T}}(\mathbf{X}^l, P_h) \in \mathbb{R}^d$, and $W_P \in \mathbb{R}^{|\mathcal{R}| \times d}$, $W_{TR} \in \mathbb{R}^{|TR| \times d}$, $W'_{TR} \in \mathbb{R}^{|TR| \times 2d}$, $W_{Len} \in \mathbb{R}^{L \times d}$, $b_P \in \mathbb{R}^{|\mathcal{R}|}$, $b_{TR}, b'_{TR} \in \mathbb{R}^{|TR|}$, $b_{Len} \in \mathbb{R}^L$ are all learnable parameters.

Given these attention vectors, the confidence of a $l$-length $Rule$ given by (1) is written as:

$$score(Rule) = f_{embdd}(w_{Len}, l) \prod_{i=1}^{l} f_{embdd}((w_P)_i^l, P_i) \, f_{embdd}((w_{TR})_{i,l+1}^l, TR_{i,l+1})$$
$$\prod_{j=1}^{l-1} \prod_{k=j+1}^{l} f_{embdd}((w_{TR})_{j,k}^l, TR_{j,k}) \tag{13}$$

where $f_{embdd}(w_{Len}, l) \in \mathbb{R}$ denotes the confidence of length $l$, $f_{embdd}((w_P)_i^l, P_i) \in \mathbb{R}$ denotes the confidence of predicate $P_i$, $f_{embdd}((w_{TR})_{i,l+1}^l, TR_{i,l+1}) \in \mathbb{R}$ denotes the confidence of temporal relation $TR_{i,l+1}$, and $f_{embdd}((w_{TR})_{j,k}^l, TR_{j,k}) \in \mathbb{R}$ denotes the confidence of temporal relation $TR_{j,k}$.

**Temporal feature modeling.** In our framework, we only focus on the connecting paths between the two entities in a query, and the TR between intervals is actually discretized, which might impair our model performance. To address these limitations, we introduce temporal features modeling where extra evidences and continuous distribution measurements are involved. Inspired by TimePlex (Jain et al. (2020)), we design an extended module. The main features considered here include $Recurrence$, $TemporalOrder$, $RelationPairInterval$ and $Duration$.

- $Recurrence$ describes the distribution of recurrence of the same relation. Different from TimePlex's description $(e_s, r, e_o, *)$, we measure this feature with a more general form $(e_s, r, *, *)$. For each relation $r$, we use a parameter $(p_{rec})_r$ to denote the probability that this relation will happen again for the same subject. It should be noted that inverse relations $r^{-1}$ are also involved.

- $TemporalOrder$ describes the distribution of the temporal order between relation pairs happening for the same subject, i.e., $(e_s, r, *, I)$, $(e_s, r', *, I')$. This feature is implied by TRs in temporal logical rules to some extent. However, for two touching intervals, we still cannot tell which one happens earlier. Thus, a parameter $(p_{order})_{r,r'}$ is adopted for every pair of relations $r$ and $r'$ to denote the probability that $r$ happens earlier than $r'$.

- $RelationPairInterval$ describes the distribution of the time gaps between relation pairs happening for the same subject. Different from TimePlex, given a pair of relations $r$ and $r'$, we consider two types of distributions for their time gap, Gaussian and exponential, with parameters $(\mu_{pair})_{r,r'}$, $(\sigma_{pair})_{r,r'}$ and $(\lambda_{pair})_{r,r'}$, respectively. Gaussian distribution is preferred when there is a roughly fixed interval such as the birth date and death date of the same person, while exponential distribution is more suitable for two strongly correlated relations.

- $Duration$ describes the distribution of the interval length of every relation. We suppose the duration of each relation $r$ follows a Gaussian distribution with parameters $(\mu_d)_r$, $(\sigma_d)_r$. It is common in large tKGs that the exact date of some facts are missing. With this feature, we can estimate these missing dates and improve our model performance.

With these temporal feature distributions, we can further evaluate the candidates of a query. Similar to TimePlex, a linear function of probability is used as scoring function, i.e., $\phi_{rec}, \phi_{order}, \phi_{pair}$. However, more evidences are used by our model in the evaluation. The evidences of TimePlex only include facts happening between the known entity and every candidate, while our model extends with the constrained random walks. Given a query $(e_s, r, ?, I)$ and a candidate $e_c$, all the evidences used in the temporal feature modeling module include facts between $e_s$ and $e_c$, i.e, $\mathcal{F}_{c,s} := \{(e_c, *, e_s, *)\}$, facts on $e_c$ but not on $e_s$, i.e., $\mathcal{F}_{c,\bar{s}} := \{(e_c, *, *, *)\} - \mathcal{F}_{c,s}$ and constrained random walks $S_{W_C}$ between $e_s$ and $e_c$. The temporal feature modeling score $\phi_{tfm}$ of the candidate is given as

$$\phi_{tfm}(e_c) = \gamma_1 \phi_{tfm,1}(e_c) + \gamma_2 \phi_{tfm,2}(e_c) + \gamma_3 \phi_{tfm,3}(e_c) \tag{14}$$

where $\phi_{tfm,1}(e_c), \phi_{tfm,2}(e_c), \phi_{tfm,3}(e_c)$ are the scoring functions given $\mathcal{F}_{c,s}, \mathcal{F}_{c,\bar{s}}$ and $S_{W_C(e_s, e_c)}$, respectively, and $\gamma_1, \gamma_2, \gamma_3 \geq 0$ are the corresponding weights. The details of these scoring functions are shown in Appendix A.

Table 1: Link prediction performance on the two benchmark datasets

| Dataset | WIKIDATA12k | | | YAGO11k | | |
|---|---|---|---|---|---|---|
| Model | MRR | hit@1 | hit@10 | MRR | hit@1 | hit@10 |
| Neural-LP | 0.1823 | 0.0908 | 0.3848 | 0.1001 | 0.0401 | 0.1845 |
| AnyBURL | 0.1908 | 0.1030 | 0.3904 | 0.0908 | 0.0378 | 0.1814 |
| TLogic | 0.2536 | 0.1754 | 0.4424 | 0.1545 | 0.1180 | 0.2309 |
| ComplEx | 0.2482 | 0.1430 | 0.4890 | 0.1814 | 0.1146 | 0.3111 |
| TA-ComplEx | 0.2278 | 0.1269 | 0.4600 | 0.1524 | 0.0936 | 0.2626 |
| HyTE (TransE) | 0.2528 | 0.1470 | 0.4826 | 0.1355 | 0.0332 | 0.2981 |
| DE-SimplE | 0.2529 | 0.1468 | 0.4905 | 0.1512 | 0.0875 | 0.2674 |
| TNT-Complex | 0.3010 | 0.1973 | 0.5069 | 0.1801 | 0.1102 | 0.3128 |
| TimePlex (base) | 0.3238 | 0.2203 | 0.5279 | 0.1835 | 0.1099 | 0.3186 |
| TimePlex | **0.3335** | 0.2278 | **0.5320** | 0.2364 | **0.1692** | 0.3671 |
| TILP (w/o tfm) | 0.3114 | 0.2152 | 0.5077 | 0.1880 | 0.1336 | 0.3089 |
| TILP | 0.3328 | **0.2342** | 0.5289 | **0.2411** | 0.1667 | **0.4149** |

**Candidate ranking.** Suppose by applying a temporal logical $Rule$, we find a total of $N$ successful random walks in which $N_c$ of them arrive at entity $e_c$. With the assumption that each walk has equal contribution, we can calculate the arriving rate $\alpha_c := N_c/N$. By aggregating all the rules learned for the target predicate, we obtain the logical rule score $\phi_{TLR}$. Combining with the temporal feature modeling score $\phi_{tfm}$ via corresponding weights $\gamma_{TLR}, \gamma_{tfm} \geq 0$, the final score $\phi_{TILP}$ becomes

$$\phi_{TLR}(e_c) = \sum_{Rule} \alpha_c(Rule)score(Rule) \tag{15}$$

$$\phi_{TILP}(e_c) = \gamma_{TLR}\phi_{TLR}(e_c) + \gamma_{tfm}\phi_{tfm}(e_c) \tag{16}$$

The training is divided into two phases. In the first phase, the attention vectors for predicates, TRs and rule length are learned by maximizing the score of correct candidates. In the second phase, all the distribution parameters of temporal features are fitted with training samples. Then we train the parameters of weights for the temporal feature modeling module with frozen attention vectors, i.e, $\phi_{TLR}$ is used for prediction in the first stage, and $\phi_{TILP}$ is adopted in the second stage.

## 6  EXPERIMENTS

### 6.1  EXPERIMENTAL SETUP

We evaluate TILP on two standard tKG datasets, WIKIDATA12k and YAGO11k (Dasgupta et al. (2018)). Detailed dataset introduction and statistics are given in Appendix B. These datasets contain temporal facts in the form $(e_s, r, e_o, I)$, e.g. (John Okell, worksAt, SOAS University of London, $[1959, 1999]$). The temporal specificity of facts in these datasets can be at the year, month, or day level, although month and day data are not present in the majority of examples; we remove month and day information from such facts to achieve a more uniform data representation. For datasets with higher granularity, we would expect improved performance due to more precise temporal relations. For the link predication task on data of the form $(e_s, r, e_o, I)$, we generate a list of ranked candidates for both object prediction $(e_s, r, ?, I)$ and subject prediction $(e_o, r^{-1}, ?, I)$. The maximum rule length is set to 5 for both datasets. The standard metrics, mean reciprocal rank (MRR), hit@1, hit@10 are used for comparison of the methods. Similar to Jain et al. (2020), we perform time-aware filtering which gives a more valid performance evaluation.

We compare TILP with state-of-the-art baselines in two dimensions: static v.s. temporal, logical-rule-based v.s. embedding-based. The static logical-rule-based methods include Neural-LP (Yang et al. (2017)), and AnyBURL (Meilicke et al. (2020)). The temporal logical-rule-based method is TLogic (Liu et al. (2021)). The static embedding-base model is ComplEx (Trouillon et al. (2016)). The temporal embedding-base model include TA-ComplEx (García-Durán et al. (2018)), HyTE

(Dasgupta et al. (2018)), DE-SimplE (Goel et al. (2019)), TNT-Complex (Lacroix et al. (2020)), and TimePlex (Jain et al. (2020)). The results for all embedding-based models are from Jain et al. (2020). Ablation studies on the temporal feature modeling module (TILP w/o tfm) are also conducted.

## 6.2 RESULTS AND ANALYSIS

The results of the experiments are shown in Table 1 with the efficiency study given in Appendix C. The performance of TILP is comparable with the best temporal embedding method across all metrics, with TimePlex performing slightly better than TILP against half of the evaluated metrics. Note that because of the human-interpretable form of the TILP predictions, its temporal logical rules provide explanatory support for predictions, while the interpretation of TimePlex embeddings would be more opaque. Some examples of the learned rules and groundings from TILP on the WIKIDATA12k dataset are given below.

*Rule 1:*

$$\text{memberOf}(E_1, E_4, I_4) \leftarrow \text{memberOf}(E_1, E_2, I_1) \wedge \text{memberOf}^{-1}(E_2, E_3, I_2)$$

$$\wedge \text{memberOf}(E_3, E_4, I_3) \wedge_{i=1}^{3} (\wedge_{j=i+1}^{4} \text{touching}(I_i, I_j))$$

*Grounding:* $E_1$ = Somalia, $E_2$ = International Development Association, $E_3$ = Kingdom of the Netherlands, $E_4$ = International Finance Corporation, $I_1$ = [1962, present], $I_2$ = [1961, present], $I_3$ = [1956, present], $I_4$ = [1962, present].

*Rule 2:*

$$\text{receiveAward}(E_1, E_4, I_4) \leftarrow \text{nominatedFor}(E_1, E_2, I_1) \wedge \text{nominatedFor}^{-1}(E_2, E_3, I_2)$$

$$\wedge \text{receiveAward}(E_3, E_4, I_3) \wedge \text{before}(I_1, I_2) \wedge_{i=1}^{2} \text{after}(I_i, I_3) \wedge_{j=1}^{3} \text{before}(I_j, I_4)$$

*Grounding:* $E_1$ = ZDF, $E_2$ = International Emmy Award for best drama series, $E_3$ = DR, $E_4$ = Peabody Awards, $I_1$ = [2005, 2005], $I_2$ = [2009, 2009], $I_3$ = [1997, 1997], $I_4$ = [2013, 2013].

**Granularity of temporal relations:** We observe that the well-known TLogic approach, which uses a point-in-time data representation of temporal facts, doesn't perform as well as more recently developed methods on these two *interval-based* tKGs. Several factors likely contribute to this phenomenon. First, the foundation of the temporal walks in TLogic is the relative order of time points. Without the introduction of temporal relation between intervals, TLogic can not be simply extended for interval-based tKGs. Second, to improve model efficiency, TLogic uses sampling strategy with the control of the total number of temporal walks. This strategy actually impairs model performance due to no guarantee for successful long-distance random walks. It is proven by the facts that a lot of temporal logical rules we found in these two datasets are of length 5, which can be challenges for TLogic.

**Temporal feature modeling:** These experiments suggest that time-aware learning is an important component for link prediction in tKGs since all static learning methods are out-performed by their counterparts with temporal learning abilities. This leaves us with a comparison of logical-rule-based methods to embedding-based methods. The former integrates temporal relations into logical rule representation, while the latter uses time-dependent embeddings. Both approaches can be implemented in a variety of ways, for example previous explicit temporal features modeling approaches have used: $Recurrence$, $TemporalOrder$, $RelationPairInterval$ and $Duration$. The alternative representation of temporal intervals as continuous distributions has demonstrated the greater success than these previous models, as shown by the evaluation of both TimePlex and TILP (our model). However, our extensions of the temporal feature modeling module are non-trivial since evidences of constrained random walks can not be found by any embedding-based methods.

## 6.3 MORE DIFFICULT PROBLEM SETTINGS

Our model uses neurally generated symbolic representations of temporal and entity relationships, while performing on par with state-of-the-art embedding-based methods. Looking beyond performance, symbolic representations convey several advantages in understanding the prediction quality in tKGs. To demonstrate these strengths, we propose the following more difficult problem settings

in consideration of data efficiency, model robustness, and transfer learning. All these scenarios are important and changeling tasks in KG reasoning, and temporal information in tKGs makes the problems even harder. A few related works (Mirtaheri et al. (2020),Xu et al. (2021),Liu et al. (2021)) try to give some solutions, but there is still room for improvements. To simplify the discussion, we restrict comparison of our model to the standard and highest performing methods for temporal link prediction: TLogic (logical-rule-based) and TimePlex (embedding-based).

**Few training samples.** Training data is often expensive to obtain for new scenarios, and therefore the performance of a method under limited training data is an important consideration. We parametrically examine the relative performance of our model in this low-data scenario, and demonstrate its data efficiency using the MRR metric. Details of the setting and the results are shown in Appendix D. It is observed that when the training set size decreases, TILP outperforms all the baseline methods. Through constrained random walks, TILP is able to capture all the patterns related to a query relation which are independent on entities. Reducing training set size only changes the frequency of different patterns. To contrast, embedding-based methods require enough training samples to learn good embeddings of entities and relations.

**Biased data.** Another common problem in knowledge graph learning is data imbalance. For some rare relations, it is hard to collect samples. For example, in the WIKIDATA12k dataset, where there are 40621 edges in total, the number of edges for relation $capitalOf$ and $residence$ are only 86 and 80, respectively. This demonstrates that achieving a good model for every relation in the dataset requires the ability to manage biased representation frequency. Details of the setting and the results are shown in Appendix D. From the results, we conclude that the attention vectors of predicates, temporal relations and rule length are relation-dependent in TILP, making it less susceptible than other methods to data imbalance. In contrast, embeddings of entities are shared among all the relations, making embedding-based method suffer more susceptible to data imbalance.

**Time shifting.** For tKG, the problem of time shifting is also noteworthy. Intuitively, more overlapping between the period of training set and that of test set brings the higher model performance. However, this condition can be violated in scenarios such as link forecasting. Similar to TLogic, we re-split the whole dataset according to the start time of each relation. Details of the setting are shown in Appendix D, and the results are shown in Table 2. One major limitation of most time-aware embedding-based methods is the use of *absolute* timestamp as anchors, preventing generalization to either time shifting settings and inductive settings (Liu et al. (2021)). With such limitations in mind, TILP extracts temporal logical rules with *relative* temporal relations, providing greater flexibility, e.g. for transfer learning to arbitrary temporal periods.

Table 2: Link prediction performance with time shifting setting

| Dataset | WIKIDATA12k | | | YAGO11k | | |
|---|---|---|---|---|---|---|
| Model | MRR | hit@1 | hit@10 | MRR | hit@1 | hit@10 |
| TimePlex (base) | 0.0975 | 0.0647 | 0.1588 | 0.0581 | 0.0299 | 0.1085 |
| TimePlex | 0.0976 | 0.0651 | 0.1580 | 0.0639 | 0.0341 | 0.1146 |
| TLogic | 0.1508 | 0.0934 | 0.3201 | 0.1107 | 0.0621 | 0.1804 |
| TILP | **0.2657** | **0.1623** | **0.4412** | **0.2069** | **0.1194** | **0.3686** |

## 7 CONCLUSION

TILP, the first differentiable framework for temporal logical rules learning, has been proposed for the link predication task on temporal knowledge graphs. Experiments on two standard datasets indicate that TILP achieves comparable performance to the state-of-the-art embedding-based methods while additionally providing logical explanations for the link predictions. In addition, we consider some important learning problems in temporal knowledge graphs, where TILP outperforms most baselines. An interesting direction for future work is to predict event intervals with temporal logical rules. In such a task, the learned rules must contain both numerical values and temporal relations, a situation which should further benefit from the expressive power of the logical rules considered in the TILP framework.

ACKNOWLEDGMENTS

This work was supported by a sponsored research award by Cisco Research.

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

## A  DETAILS OF TEMPORAL FEATURE MODELING

Given a query $(e_s, r, ?, I)$ and a candidate $e_c$, the facts used in this module include three parts: the set of edges from $e_c$ to $e_s$ denoted by $\mathcal{F}_{c,s}$, the set of edges from $e_c$ to other entities denoted by $\mathcal{F}_{c,\bar{s}}$, and the set of paths from $e_s$ to $e_c$ denoted by $S_{W_C(e_s,e_c)}$. Let $\mathcal{R}^{(1)}, \mathcal{R}^{(2)}, \mathcal{R}^{(3)} \subseteq \mathcal{R}$ denote the set of relations existing in $\mathcal{F}_{c,s}$, $\mathcal{F}_{c,\bar{s}}$ and $S_{W_C(e_s,e_c)}$, respectively. Given a relation $r'$, let $(T_s)^1_{r'}$, $(T_s)^2_{r'}$, $(T_s)^3_{r'} \in \mathbb{R}$ be the closest start time, existing in $\mathcal{F}_{c,s}$, $\mathcal{F}_{c,\bar{s}}$ and $S_{W_C(e_s,e_c)}$, to the start time of the query interval denoted by $t_s$, respectively. In addition, we introduce a scoring function $\phi$ to integrate different probabilities:

$$\phi(e_c; h, w, b) = \frac{\sum_{r' \in \mathcal{R}(e_c)} \exp(w_{r,r'})(h_{r,r'} + b_{r,r'})}{\sum_{r'' \in \mathcal{R}(e_c)} \exp(w_{r,r''})} \tag{17}$$

where $\phi(e_c) \in \mathbb{R}$ denotes the score of $e_c$, the probability $h \in \mathbb{R}^{|\mathcal{R}| \times |\mathcal{R}|}$ is corresponding with the query relation $r$ and another relation $r'$, i.e., $h_{r,r'} \in \mathbb{R}$, $\mathcal{R}(e_c) \subseteq \mathcal{R}$ denotes the set of relations related to $e_c$, and $w, b \in \mathbb{R}^{|\mathcal{R}| \times |\mathcal{R}|}$ are learnable parameters, i.e, $w_{r,r'}, b_{r,r'} \in \mathbb{R}$.

In our model, given a temporal feature $x \in \mathbb{R}$ related to candidate $e_c$, query relation $r$, and another relation $r'$, its probability $h_{r,r'}$ may follow three types of distributions: 1) Bernoulli with parameter $p \in \mathbb{R}^{|\mathcal{R}| \times |\mathcal{R}|}$, i.e., $h_{r,r'} = (p_{r,r'})^x (1 - p_{r,r'})^{1-x}$; 2) Gaussian with parameters $\mu, \sigma \in \mathbb{R}^{|\mathcal{R}| \times |\mathcal{R}|}$, i.e., $h_{r,r'} = \mathcal{N}(x; \mu_{r,r'}, \sigma_{r,r'})$; 3) Exponential with parameter $\lambda \in \mathbb{R}^{|\mathcal{R}| \times |\mathcal{R}|}$, i.e., $h_{r,r'} = \lambda_{r,r'} \exp(-\lambda_{r,r'} x)$.

For example, with a yearly resolution in YAGO11k dataset, we found the $RelationPairInterval$ between a person's birth and graduation is normally distributed with mean 22 and standard deviation 1, and that between a person's birth and death is also normally distributed with mean 70 and standard deviation 6. Let $r = $ wasBornIn, $r' = $ graduatedFrom, $r'' = $ diedIn, we have $\mu_{r,r'} = 22$, $\sigma_{r,r'} = 1$, $\mu_{r,r''} = 70$, $\sigma_{r,r''} = 6$. Given a query $(?, \text{wasBornIn}, \text{Nashville}, [1872, 1872])$ and a candidate $e_c = $ Cass Canfield, we first check the known facts related to $e_c$ which include (Cass Canfield, graduatedFrom, Harvard University, $[1919, 1919]$) and (Cass Canfield, diedIn, New York City, $[1986, 1986]$). Then the temporal feature, i.e., $RelationPairInterval$, related to $e_c$, $r$ and $r'$ is $x = |1872 - 1919| = 47$. Thus, $h_{r,r'} = \mathcal{N}(x; \mu_{r,r'}, \sigma_{r,r'}) = \mathcal{N}(47; 22, 1) = 7.65 \times e^{-137}$. Similarly, $h_{r,r''} = \mathcal{N}(|1872 - 1986|; 70, 6) = 1.39 \times e^{-13}$. These probabilities $h_{r,r'}, h_{r,r''}$ are integrated with (17) to form the scoring function $\phi_{pair}(e_c)$. As for other temporal features, the calculation processes of the scoring functions are similar. The differences are: 1) Since $Recurrence$ is only related to the query relation, we use (18) which can be considered as a simplified version of (17). 2) Both $Recurrence$ and $TemporalOrder$ follow Bernoulli distribution, i.e., $x = 0$ or 1. More details are shown below.

- $Recurrence$ describes the probability distribution of recurrence of relation $r$ on entity $e_c$, and is considered in $\mathcal{F}_{c,s}$ and $\mathcal{F}_{c,\bar{s}}$. We suppose this probability follows a Bernoulli distribution with parameter $p_{rec,1}, p_{rec,2} \in \mathbb{R}^{|\mathcal{R}|}$ in $\mathcal{F}_{c,s}$ and $\mathcal{F}_{c,\bar{s}}$, respectively. The following scoring function is defined:

$$\phi_{rec}(e_c; h_{rec}, w_{rec}, b_{rec}) = (w_{rec})_r (h_{rec})_r + (b_{rec})_r \tag{18}$$

  where the temporal feature related to $e_c$ and $r$ is $x = \mathbb{1}(r \in \mathcal{R}(e_c))$, its probability $(h_{rec})_r = ((p_{rec})_r)^x (1 - (p_{rec})_r)^{1-x} \in \mathbb{R}$ is based on $p_{rec} \in \mathbb{R}^{|\mathcal{R}|}$, i.e., $h_{rec} \in \mathbb{R}^{|\mathcal{R}|}$, and $w_{rec}, b_{rec} \in \mathbb{R}^{|\mathcal{R}|}$ are learnable parameters. Given $\mathcal{R}(e_c) = \mathcal{R}^{(1)}$, we have $\phi_{rec,1}(e_c; h_{rec,1}, w_{rec,1}, b_{rec,1})$ with parameters $h_{rec,1}, w_{rec,1}, b_{rec,1} \in \mathbb{R}^{|\mathcal{R}|}$; Given $\mathcal{R}(e_c) = \mathcal{R}^{(2)}$, we have $\phi_{rec,2}(e_c; h_{rec,2}, w_{rec,2}, b_{rec,2})$ with parameters $h_{rec,2}, w_{rec,2}, b_{rec,2} \in \mathbb{R}^{|\mathcal{R}|}$.

- $TemporalOrder$ describes the probability distribution of the temporal order of two relations $r$ and $r'$, and is considered in $\mathcal{F}_{c,s}$, $\mathcal{F}_{c,\bar{s}}$ and $S_{W_C(e_s, e_c)}$. We suppose this probability follows a Bernoulli distribution with parameter $p_{order,1}, p_{order,2}, p_{order,3} \in \mathbb{R}^{|\mathcal{R}| \times |\mathcal{R}|}$ in $\mathcal{F}_{c,s}$, $\mathcal{F}_{c,\bar{s}}$ and $S_{W_C(e_s, e_c)}$, respectively. Given $\mathcal{R}(e_c) = \mathcal{R}^{(1)}$, we calculate $\phi_{order,1}(e_c; h_{order,1}, w_{order,1}, b_{order,1})$ with (17), where $h_{order,1} \in \mathbb{R}^{|\mathcal{R}| \times |\mathcal{R}|}$ is based on $p_{order,1}$ with $x = \mathbb{1}(t_s < (T_s)_{r'}^1)$, and $w_{order,1}, b_{order,1} \in \mathbb{R}^{|\mathcal{R}| \times |\mathcal{R}|}$ are learnable parameters. Similarly, given $\mathcal{R}(e_c) = \mathcal{R}^{(2)}$, we calculate $\phi_{order,2}(e_c; h_{order,2}, w_{order,2}, b_{order,2})$ with parameters $h_{order,2}, w_{order,2}, b_{order,2} \in \mathbb{R}^{|\mathcal{R}| \times |\mathcal{R}|}$, where $h_{order,2}$ is based on $p_{order,2}$ with $x = \mathbb{1}(t_s < (T_s)_{r'}^2)$; Given $\mathcal{R}(e_c) = \mathcal{R}^{(3)}$, we calculate $\phi_{order,3}(e_c; h_{order,3}, w_{order,3}, b_{order,3})$ with parameters $h_{order,3}, w_{order,3}, b_{order,3} \in \mathbb{R}^{|\mathcal{R}| \times |\mathcal{R}|}$, where $h_{order,3}$ is based on $p_{order,3}$ with $x = \mathbb{1}(t_s < (T_s)_{r'}^3)$.

- $RelationPairInterval$ describes the distribution of the time gap between two relations $r$ and $r'$, and is considered in $\mathcal{F}_{c,s}$, $\mathcal{F}_{c,\bar{s}}$ and $S_{W_C(e_s, e_c)}$. We suppose this probability follows a Gaussian or exponential distribution with parameters $\mu_{pair,1}, \sigma_{pair,1}, \lambda_{pair,1}$, $\mu_{pair,2}, \sigma_{pair,2}, \lambda_{pair,2}, \mu_{pair,3}, \sigma_{pair,3}, \lambda_{pair,3} \in \mathbb{R}^{|\mathcal{R}| \times |\mathcal{R}|}$ in $\mathcal{F}_{c,s}$, $\mathcal{F}_{c,\bar{s}}$ and $S_{W_C(e_s, e_c)}$, respectively. Given $\mathcal{R}(e_c) = \mathcal{R}^{(1)}$, we calculate $\phi_{pair,1}(e_c; h_{pair,1}, w_{pair,1}, b_{pair,1})$

with (17), where $h_{pair,1} \in \mathbb{R}^{|\mathcal{R}| \times |\mathcal{R}|}$ is based on $\{\mu_{pair,1}, \sigma_{pair,1}\}$ or $\lambda_{pair,1}$ with $x = |t_s - (T_s)^1_{r'}|$, and $w_{pair,1}, b_{pair,1} \in \mathbb{R}^{|\mathcal{R}| \times |\mathcal{R}|}$ are learnable parameters. Similarly, given $\mathcal{R}(e_c) = \mathcal{R}^{(2)}$, we calculate $\phi_{pair,2}(e_c; h_{pair,2}, w_{pair,2}, b_{pair,2})$ with parameters $h_{pair,2}, w_{pair,2}, b_{pair,2} \in \mathbb{R}^{|\mathcal{R}| \times |\mathcal{R}|}$, where $h_{pair,2}$ is based on $\{\mu_{pair,2}, \sigma_{pair,2}\}$ or $\lambda_{pair,2}$ with $x = |t_s - (T_s)^2_{r'}|$; Given $\mathcal{R}(e_c) = \mathcal{R}^{(3)}$, we calculate $\phi_{pair,3}(e_c; h_{pair,3}, w_{pair,3}, b_{pair,3})$ with parameters $h_{pair,3}, w_{pair,3}, b_{pair,3} \in \mathbb{R}^{|\mathcal{R}| \times |\mathcal{R}|}$, where $h_{pair,3}$ is based on $\{\mu_{pair,3}, \sigma_{pair,3}\}$ or $\lambda_{pair,3}$ with $x = |t_s - (T_s)^3_{r'}|$.

- *Duration* describes the probability distribution of interval length of every relation. We suppose this probability follows a truncated Gaussian distribution with parameters $\mu_d, \sigma_d \in \mathbb{R}^{|\mathcal{R}|}$ within the interval $[0, +\infty)$. It is common in large tKGs that the exact date of some events are missing. With this feature, we can estimate these missing dates and improve our model performance. For example, given a fact or query with incomplete interval $I = [t_s, ?]$, we generate a duration $t_d \sim \psi((\mu_d)_r, (\sigma_d)_r, 0, +\infty)$ where $r$ is the corresponding relation, and set $\hat{I} = [t_s, t_s + t_d]$.

With these temporal feature distributions, we can further evaluate the candidates given a query. We fisrt combine the scores in each part with (19)-(21). Then the scores from different parts are combined to obtain the temporal feature modeling score $\phi_{tfm}$ given in (14).

$$\phi_{tfm,1}(e_c) = \gamma_{rec,1}\phi_{rec,1}(e_c) + \gamma_{order,1}\phi_{order,1}(e_c) + \gamma_{pair,1}\phi_{pair,1}(e_c) \tag{19}$$

$$\phi_{tfm,2}(e_c) = \gamma_{rec,2}\phi_{rec,2}(e_c) + \gamma_{order,2}\phi_{order,2}(e_c) + \gamma_{pair,2}\phi_{pair,2}(e_c) \tag{20}$$

$$\phi_{tfm,3}(e_c) = \gamma_{order,3}\phi_{order,3}(e_c) + \gamma_{pair,3}\phi_{pair,3}(e_c) \tag{21}$$

where all $\gamma \geq 0$ are learnable weights. For each part, the sum of weights is equal to 1, i.e., $\gamma_{rec,1} + \gamma_{order,1} + \gamma_{pair,1} = 1$, $\gamma_{rec,2} + \gamma_{order,2} + \gamma_{pair,2} = 1$, and $\gamma_{order,3} + \gamma_{pair,3} = 1$.

## B  DATASET DISCUSSION

There are four benchmark temporal knowledge graph datasets including ICEWS(Lautenschlager et al. (2015)), GDELT(Leetaru & Schrodt (2013)), WIKIDATA(Leblay & Chekol (2018)) and YAGO (Mahdisoltani et al. (2014)). As for the temporal representation, the first two datasets use timestamps, while the last two uses intervals. Compared with timestamp-based methods, interval-based tKGs are both more general and more difficult to learn. Thus, we mainly focus on WIKIDATA and YAGO datasets in our experiments.

WIKIDATA is a large knowledge base based on Wikipedia. To form the WIKIDATA12k dataset, a subgraph with temporal information is extracted by Dasgupta et al. (2018). It is guaranteed that every single node are related to multiple facts, and the top 24 frequent relations are selected. YAGO is another large knowledge graph built from multilingual Wikipedias. Similarly, some temporally associated facts are distilled out from YAGO3 to form the YAGO11k dataset (Dasgupta et al. (2018)). In this dataset, every single node is connected by more than one edge, and the top 10 frequent relations are selected. For the WIKIDATA12k and YAGO11k datasets, they contain many time-sensitive relations such as 'residence', 'position held', 'member of sports team', 'member of', 'educated at' in WIKIDATA12k and 'worksAt', 'playsFor', 'isAffiliatedTo', 'hasWonPrize', 'owns' in YAGO11k. These time-sensitive relations make the link prediction task in these two datasets more challenging. For example, in the WIKIDATA12k datatset, a person can become a member of different teams, hold different positions, and receive different awards in various periods. Thus, it is necessary to model temporal information for link prediction tasks in tKGs. Table 3 shows the dataset statistics with a yearly time resolution, where $\mathcal{G}, \mathcal{E}, \mathcal{R}, \mathcal{T}, \mathcal{I}$ are the set of edges, entities, relations, timestamps and intervals, respectively.

## C  EFFICIENCY STUDY

For the rule learning process, the time complexity of TLR module (path search, rule extraction, and the arriving rate calculation) is $\mathcal{O}(N_{pos}N_{rule}(L|\mathcal{G}| + L^2 N_{path}))$, where $L$ is the maximum rule length, $N_{pos}$ is the number of positive examples, $N_{rule}$ is the maximum number of rules found in

Table 3: Dataset statistics with a yearly time resolution

| Dataset | $\lvert\mathcal{G}_{train}\rvert$ | $\lvert\mathcal{G}_{valid}\rvert$ | $\lvert\mathcal{G}_{test}\rvert$ | $\lvert\mathcal{E}\rvert$ | $\lvert\mathcal{R}\rvert$ | $\lvert\mathcal{T}\rvert$ | $\lvert\mathcal{I}\rvert$ |
|---|---|---|---|---|---|---|---|
| WIKIDATA12k | 32,497 | 4,062 | 4,062 | 12,544 | 24 | 237 | 2,564 |
| YAGO11k | 16,408 | 2,051 | 2,050 | 10,622 | 10 | 251 | 6,651 |

a single example, and $N_{path}$ is the maximum number of paths for a given rule that can be found in a single example. The time complexity of tfm module (distribution parameters measurement) is $\mathcal{O}(N_{pos}(\delta + \lvert\mathcal{R}\rvert + LN_{rule}N_{path}))$, where $\delta$ is the maximum node degree. Similarly, for the rule application process, the time complexity of TLR module is $\mathcal{O}(N_{qry}N'_{rule}(L\lvert\mathcal{G}\rvert + L^2 N'_{path}))$, where $N_{qry}$ is the number of queries, $N'_{rule}$ is the maximum number of rules for a given target predicate, and $N'_{path}$ is the maximum number of paths for a given rule that can be found in a single query. The time complexity of tfm module is $\mathcal{O}(N_{qry}(K\delta + K\lvert\mathcal{R}\rvert + LN'_{rule}N'_{path}))$, where $K$ is the maximum number of candidates for a single query. Since the path search for each positive example and the rule application for each query are both independent, these processes are done in parallel. Given the maximum rule length $L = 5$, the rule searching of TLR module on a 4-CPU machine takes 1740.6s on WIKIDATA12k training set, and 571.9s on YAGO11k training set. The distribution parameters measurement of tfm module takes 44.1s on WIKIDATA12k training set, and 6.7s on YAGO11k training set. The rule application of TLR module takes 1522.8s on WIKIDATA12k validation set, and 529.6s on YAGO11k validation set. The scoring of tfm module takes 1713.8s on WIKIDATA12k validation set, and 527.8s on YAGO11k validation set.

## D  DETAILS OF THE MORE DIFFICULT PROBLEM SETTINGS

**Few training samples.** In experiments, we randomly reduce the number of training samples in training set and evaluate different models for the two datasets. To alleviate the effects of different data distributions, we repeat this experiments for 5 rounds. The results are shown in Fig.2, where we draw the average MRR curve with err bars. When the training set size decreases, TILP outperforms all the baseline methods. Through constrained random walks, TILP is able to capture all the patterns related to a query relation which are independent on entities. Reducing training set size only changes the frequency of different patterns. To contrast, embedding-based methods require enough training samples to learn good embeddings of entities and relations. In this setting, our method is also better than TLogic, which demonstrates the advantages of the neural-network-based logical rule learning framework relative to the statistical methods.

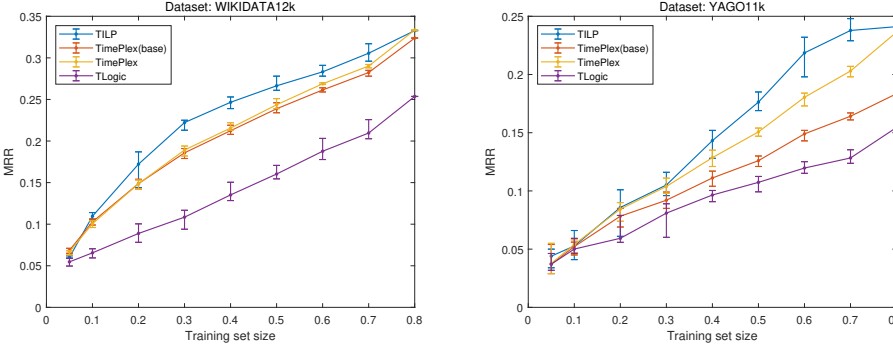

Figure 2: Link prediction performance with few training samples

**Biased data.** For the two datasets, we first obtain statistic results on the number of edges for each relation. Then for a specific relation, we randomly reduce $50\%$ of the edges in the training set, and evaluate the performance changes caused by this setting. We conduct the experiments for every relation separately, and repeat for 5 rounds to alleviate the influence of different data distributions.

The original test set from random generation contains few queries of rare relations. To guarantee enough test queries for each relation, we adjust the distribution of queries in the test set, trying to set the number of queries of different relations to be equal. If queries of a certain relation are not enough, we would randomly choose half of them. In Fig.3, the average change of MRR for different models is shown, where error bars have been suppressed for readability. We conclude that the attention vectors of predicates, temporal relations and rule length are relation-dependent in TILP, making it less susceptible than other methods to data imbalance. In contrast, embeddings of entities are shared among all the relations, making embedding-based method suffer more susceptible to data imbalance. As a statistical method, TLogic also fails since it can not build dependency between different rules.

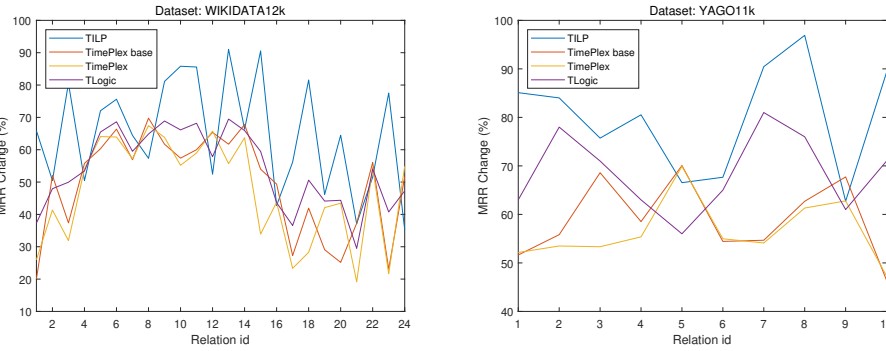

Figure 3: Link prediction performance with biased data

**Time shifting.** In practice, we put the edges missing start time into training set, and correct the edges with wrong start time (greater than the year of 2022). For WIKIDATA12k dataset, the start time range for training set, validation set and test set are $[0, 2008]$, $[2008, 2012]$, $[2012, 2018]$, respectively. For YAGO11k dataset, the start time range for training set, validation set and test set are $[-431, 2006]$, $[2006, 2011]$, $[2011, 2022]$, respectively. The results are shown in Table 2. One major limitation of most time-aware embedding-based methods is the use of *absolute* timestamp as anchors, preventing generalization to either time shifting settings and inductive settings (Liu et al. (2021)). With such limitations in mind, TILP extracts temporal logical rules with *relative* temporal relations, providing greater flexibility, e.g. for transfer learning to arbitrary temporal periods. In this setting, our method is still better than TLogic which builds their model on timestamps, and ignores the necessity of learning all possible temporal patterns from data.

