# OpenReview forum: "TILP: Differentiable Learning of Temporal Logical Rules on Knowledge Graphs"
_ICLR.cc/2023/Conference — ICLR 2023 poster_

### Official Review · Reviewer_qZrk · 2022-10-24

**Confidence:** 2
**Correctness:** 2
**Technical Novelty And Significance:** 2
**Empirical Novelty And Significance:** 2
**Recommendation:** 5

**Clarity, Quality, Novelty And Reproducibility:**

The clarity is bad for me, but may due to the lack of background.
Quality is fine.
The method and studying scenario is novel, however I am not sure if those scenarios are interesting due to the lack of background.

**Strength And Weaknesses:**

## Strength

- This method proposes a diiferentiable logic rule method on knowledge graph which can deal with time range related rules which has no previous works.

## Weakness

A lot of notations and explanations need to be refined for clarity.

-  In section 4, why call the first kind Markovian? I think Markovian refers to the case that current state is only determined by previous state. However, in your case, current random walk step is not determined by previous random walk step, instead, it is determined by a given relation type and given starting time which never changes in a random walk.
- Isn't Eq (6) wrong? In the left side, v is subscripted by length, while in the right side, v is subscripted by node identity.
- You should specify the dimension of each variable and function for Eq (8-13). It is very hard for me to understand the process there.
- What is $h_{0, l}$ in Eq (8)? Simply 0 as common RNN?
- Why this random walk embedding  Eq (8-12) only cares the context relation $r_h$, and ignores relations in the random walk?
- Can you provide the exact computation for all four temporal feature modeling? The definition is too vague for me to confidently understand.
- I do not see the usage of recurrence and duration in Eq (14-17).
- In section 5, it says that you will use all possible pathes between two nodes for both training and inference. So there is no "random" walk. Isn't this very costy?

Empirical result is not satisfying.

- On only two datasets, TILP does not show clearly better performance than TimePlex. Given the fact that this method is not incremental (does not need to store full history of temporal graph), this performance is not impressive.
- Although this paper shows TILP has the potential in restricted scenario, I am not sure if those hypothesis is interesting in tKG field or has no previous works, since I am unfamiliar with this field.
- It is unclear why TLogic works extremely worse since time range logics can be expressed by timestamp logics, thus I will expect similar performance.

**Summary Of The Paper:**

This paper proposes a differentiable temporal logical rule method, TILP for temporal knowledge graph. The proposal can achieve similar performance as state-of-the-art baselines, while can find helpful logic patterns under some restricted scenario which no previous work can find.

**Summary Of The Review:**

This paper proposes a differentiable temporal logical rule method for temporal knowledge graph which has no previous works. However, the empirical gain is limited compare with existing baselines. The potential of restricted learning is not persuasive for me since those restricted scenario is only on two datasets, and no related works are provided to show community interest in those scenario.

---

> ### Author Response · Authors · 2022-11-16
> **Response to Reviewer qZrk**
>
> Thanks for the comments. In the following we answer the raised issues.
>
> -**Markovian property in random walk**
> Let $v_i\in[0,1]^{|\mathcal{E}|}$ be the state probability for step $i\in \mathbb{N}$ where $\mathcal{E}$ is the set of entities. This state probability $v_i$ is to describe the probability that we arrive at different entities after $i-1$ steps of random walk, i.e., $\sum_{x=1}^{|\mathcal{E}|} {(v_{i})}_{x} =1$.
>
> For unconstrained random walk or constrained random walk under Markovian constraints, the next state probability can be only determined by current state probability. Thus, we can use matrix multiplication to represent the random walk process, i.e., $v_{i+1}=Mv_{i}$ where $M$ is the matrix operator. Note that in the paper $v_i$ is unnormalized, but we can always normalize it by $\frac{v_i}{\sum_{x=1}^{|\mathcal{E}|}(v_i)_x}$.
>
> To contrast, for constrained random walk under non-Markovian constraints, $v_{i+1}$ is determined by all the previous vectors {$v_j$} for $j\leq i$. Thus, we can not simply use matrix multiplication to represent the random walk process. Instead, we check each path to ensure these non-Markovian constraints are satisfied.
>
> -**Clarity of Eq.(6)**
> Eq.(6) has been rewritten such that for both sides $v$'s subscript is the index of the step in the walk.
>
> -**Clarity of Eq.(8)-(13)**
> The dimension of each variable and function has been specified in the revision.
>
> -**$h_{0,l}$ in Eq.(8)**
> We set $h_{0,l}$ as $\mathbf{0}$ similar to common RNN.
>
> -**Random walk embedding in Eq.(8)-(12)**
> Given the target predicate $P_h$, Eq.(8)-(12) are used to solve the attention vectors of different constraints, e.g., $({w_P})^l_i\in\mathbb{R}^{|\mathcal{R}|}$ denotes the attention vector for the $i$-th step predicate in the $l$-length rule, where $\mathcal{R}$ is the set of relations. In Eq.(8)-(12), the exact predicates in a rule are not involved since these attention vectors are shared among different rules. When given a specific rule extracted from random walk, we calculate its score with Eq.(13) where the relations in random walk are involved. For example, if a $l$-length rule includes $r_2$ at step 1, and $r_7$ at step 2, its score would be related to $f_{embdd}(({w_P})^l_1, r_2)$ and $f_{embdd}(({w_P})^l_2, r_7)$.
>
> -**Exact computation for all four temporal features modeling**
> We add this part in Appendix A of the revision.
>
> -**Usage of recurrence and duration**
> Given a candidate entity $e_c$, $Recurrence$ describes the probability distribution of recurrence of the target predicate. Thus, this feature is considered in $\mathcal{F}{c,s}$ and $\mathcal{F}{c,\bar{s}}$. As for $Duration$, we mainly use it to estimate the incomplete intervals in our datasets. Please check Appendix A of the revision for more details.
>
> -**Using all possible paths**
> Random walk is a tradeoff between cost and accuracy, i.e., by using sampling, the cost can be controlled but the accuracy will also drop. For our datasets, both of them are sparse tKGs (see Appendix B of the revision). Besides, we limit the maximum rule length to 5, so the complexity is kept in bound.
>
> -**Empirical results**
> I) Although TILP does not show significant performance improvements than TimePlex, our model does have quite a few advantages. Given a query, our model can provide not only a ranked list of candidates, but also some logical rules for explanation. These logical rules are useful representations compatible with both machine learning models and human knowledge. The ever-increasing attentions on Explainable AI and Ethics of AI show the necessity of introducing logical learning into the area of ML.
>
> II) All these scenarios are important and changeling tasks in KG reasoning, and temporal information in tKGs makes the problems even harder. Most previous works in tKGs ignore these scenarios since embedding-based methods can hardly handle them. There are a few related works which we cited in Section 6.3 of the revision. However, there is still room for improvements.
>
> III) TLogic has these limitations: First, its definition of temporal relations is based on timestamp. Given the start time of two facts, you can only judge which event happened earlier, but do not know whether the events overlap or not. In real world, overlapping means co-existence and interaction, which are important for logical learning. Second, the temporal relations in TLogic are fixed, i.e., it does not have the ability to learn temporal patterns. If a rule has different temporal relations from the fixed ones, it can not be learned by TLogic. Third, to reduce the cost, TLogic uses some sampling strategy which impairs its performance especially when the rule length is long.
>
> IV) The benchmark tKG datasets include ICEWS, GDELT, WIKIDATA and YAGO. We focus on WIKIDATA12k and YAGO11k since they are interval-based datasets. We discuss these datasets in Appendix B of the revision.

---

### Official Review · Reviewer_EsnR · 2022-10-24

**Confidence:** 3
**Clarity, Quality, Novelty And Reproducibility:** see above
**Correctness:** 3
**Technical Novelty And Significance:** 3
**Empirical Novelty And Significance:** 2
**Recommendation:** 6

**Strength And Weaknesses:**

Strengths:
1. The motivation is reasonable.
2. The paper proposes a novel framework to model various temporal logical rules.
3. The experiments demonstrates the effectiveness of the proposed model.

Weaknesses:

The presentation needs further improvements and the paper is difficult to follow. For example:

 1) what is the difference between r and P?
 2) The notations in Section 4 are not very clear, why sometimes using h and t for head and tail entities, and sometimes using x and y?
 3) What does the operator M mean? An example may be helpful.
 4) The task formulation is actually similar with conventional link prediction without time interval. And, in table 1, why the ComplEx performs much better than TA-ComplEx, the temporal version of it? I would suggest an explanation of the dataset, so that we can better understand the necessity of modeling temporal information for link prediction.

**Summary Of The Paper:**


The paper proposes a novel method to model temporal logical rules for knowledge graph completion. In specific, the authors first define constrained random walk for rule learning. Then, based on learned rules, the authors design a module to model various temporal features and apply rules for prediction.
================================================================
I have read the response and that solved most of my concerns. I would like to raise my score to 6.

**Summary Of The Review:**

see above

---

> ### Author Response · Authors · 2022-11-16
> **Response to Reviewer EsnR**
>
> Thanks for the comments which help us improve our paper. In the following we answer the raised issues. For the revision, we proofread the paper and polish it’s presentation entirely.
>
> -**Difference between r and P**
>
> $r$ is a relation of facts in knowledge graphs, while $P$ is a predicate in temporal logical rules. Here the predicate is a fixed relation in rules, meaning that these predicates will not change in rule application. The difference is that, given a knowledge graph, the set of relations $\mathcal{R}$ and the index of its members are fixed, while we learn $P\in\mathcal{R}$ from data, and use the step of the random walk as its index. For example, in a certain rule, we can say $P_1=r_{10}$ and $P_2=r_3$ where $P_1$ is the predicate of the first step of the walk, $P_2$ is the predicate of the second step of the walk, and $r_3,r_{10}$ are the two relations existing in our database.
>
> -**Notations in Section 4**
>
> Notations in Section 4 are rewritten in the revision for clarity. In the paper, $e_s$ and $e_o$ denote the subject and object entity of a fact, respectively. When constructing the matrix operator, we consider every pair of entities $e_x,e_y\in\mathcal{E}$, where $\mathcal{E}$ is the set of entities, as intermediate entities. Here, $M_{x,y}$ denotes the $(x,y)$ entry of the matrix operator $M$. Thus, when we check all entries of $M$, we actually consider all pairs of entities as intermediate entities.
>
> -**Meaning of the operator**
>
> The matrix operator $M_{CM} \in${$0,1$}$^{|\mathcal{E}| \times|\mathcal{E}|}$, where $\mathcal{E}$ is the set of entities, is defined as follows: its $(x, y)$ entry denoted by $\left(M_{CM}\right)_{x,y}$ is 1 iff there exists at least one edge from $e_y$ to $e_x$ that satisfies the constraint $CM$. The essence of the operator is the adjacency matrix under Markovian constraints, and we set the entry maximum to 1.
>
> For example, given a simple tKG where $e_1$ = David Beckham, $e_2$ = Real Madrid, $e_3$ = AC Milan, $r_1$ = plays for, $r_2$ = cooperate with, there exist four facts in total: $(e_1,r_1, e_2, [2003,2007])$, $(e_1, r_1, e_3,[2008,2009])$, $(e_2,r_2,e_3,[2005,2010])$, and $(e_3,r_2,e_2,[2005,2010])$. Let the constraint $CM$ = \{plays for, before($I$, $[2008,2009]$)\}, for the operator $M_{CM}\in${$0,1$}$^{3\times3}$, we have $(M_{CM})_{2,1}=1$, and all the other entries are 0.
>
> If we start from $e_1$, using matrix multiplication $v_2 = M_{CM}v_1$ where $v_1=[1,0,0]^T$ is the indicator vector for start,  we will obtain $v_2=[0,1,0]^T$ which is the indicator vector after one step of random walk, i.e, we arrive at $e_2$ since there exists at least one edge from $e_1$ to $e_2$ that satisfies $CM$.
>
> -**Task formulation**
>
> I) In real world, many facts change over time, e.g., something that is true in 2020, may not be true anymore in 2022. Ignoring such temporal information may lead to ambiguity and incorrect conclusions. Thus, the introduction of temporal information in knowledge graphs is necessary. Besides, when predicting links in tKGs, it becomes much more difficult: Given a query with the same subject entity and relation, the correct object entity can change with different query intervals. For example, for the query (David Beckham, plays for, ?, [2003,2007]), the answer should be Real Madrid; but for the query (David Beckham, plays for, ?, [2008,2009]), the answer should be AC Milan.
>
> II) For the ComplEx and TA-ComplEx model, we use the performance reported by the authors of TimePlex[1]. The TA-ComplEx model is composed of two parts: the TA-module is utilized to learn time-aware representations, which are incorporated into the ComplEx framework. There might be some compatibility issues for the two methods. In the paper of TA-module[2], the authors combine TA-module with either TransE or distMult, i.e., TA-TransE and TA-distMult, which might be more suitable.
>
> III) We add the discussion of datasets in Appendix B of the revision. For WIKIDATA12k and YAGO11k datasets, they contain many time-sensitive relations such as 'residence', 'position held', 'member of sports team', 'member of', 'educated at' in WIKIDATA12k and 'worksAt', 'playsFor', 'isAffiliatedTo', 'hasWonPrize', 'owns' in YAGO11k. These time-sensitive relations make the link prediction task in these two datasets more challenging. For example, in the WIKIDATA12k datatset, a person can become a member of different teams, hold different positions, and receive different awards in various periods. Thus, it is necessary to model temporal information for link prediction tasks in tKGs.
>
> Reference:
>
> [1] Jain, Prachi, Sushant Rathi, and Soumen Chakrabarti. "Temporal knowledge base completion: New algorithms and evaluation protocols." arXiv preprint arXiv:2005.05035 (2020).
>
> [2] García-Durán, Alberto, Sebastijan Dumančić, and Mathias Niepert. "Learning sequence encoders for temporal knowledge graph completion." arXiv preprint arXiv:1809.03202 (2018).

---

### Official Review · Reviewer_HUCP · 2022-10-24

**Confidence:** 3
**Clarity, Quality, Novelty And Reproducibility:** The codes are not provided.
**Correctness:** 2
**Technical Novelty And Significance:** 2
**Empirical Novelty And Significance:** 2
**Recommendation:** 5

**Strength And Weaknesses:**

### Strength
1. The problem of learning temporal logical rules is important.
2. It improves the random walk in existing works from based on timestamps into based on intervals, which more flexible.

### Weakness
1. The authors mention that StreamLearner and TLogic are both statistic learning methods, but what are the drawbacks of such methods? What type is the authors' method and what are the advantage?
2. The writing needs improvement, for example, it should emphasize in the introduction more the reason why this model is better than other TRL methods such as [1].
3. The experiments are not adequate, lacking efficiency study.

[1] Liu Y, Ma Y, Hildebrandt M, et al. Tlogic: Temporal logical rules for explainable link forecasting on temporal knowledge graphs[C]//Proceedings of the AAAI Conference on Artificial Intelligence. 2022, 36(4): 4120-4127.

**Summary Of The Paper:**

This paper proposes a method for learning temporal logical rules. Temporal logical rules are naturally discrete, this paper utilizes a differentiable framework for learning such rules. Specifically, it proposes a random walk mechanism and designs several operators for the temporal relations. It conducts experiments on several real-world datasets to demonstrate its effectiveness.

**Summary Of The Review:**

The problem of TRL is crucial, but the core advantage of this method is unclear; furthermore, I have some concerns about reproducibility due to the lack of codes.

---

> ### Author Response · Authors · 2022-11-17
> **Response to Reviewer HUCP**
>
> We appreciate the reviewer's comments, which help us improve our paper. In the following we answer the raised issues. For the revision, we proofread the paper and polish it’s presentation entirely. Please check the revision for more details.
>
> -**Drawbacks of statistical methods**
>
> First, these statistical methods count from graph the number of paths that support a given rule as its confidence estimation. As such, this independent rule learning ignores the interactions between different rules from the same positive example. For instance, given certain rules, the confidence of some rules might be enhanced, while that of others can be diminished. Second, these methods cannot deal with the similarity between different rules. Given a reliable rule, it is reasonable to believe that the confidence of another similar rule, e.g., with the same predicates but slightly different temporal patterns, is also high. However, its estimated confidence with these methods can be quite low if it is infrequent in the dataset. Finally, the performance of these timestamp-based methods on interval-based tKGs is not demonstrated. It should be noted that the temporal relations between intervals are more complex than those of timestamps.
>
> Alternatively, our method is a differentiable neural-network-based method, which has the following advantages: I) It considers the interactions between different rules from the same positive example. The loss function of our neural network is the score of the correct entity which aggregates all the rules related to this entity. II) Our framework can deal with the similarity between different rules since the attention vectors are shared among different rules. If two rules are similar, the embedding lookup function will fetch similar values according the predicates and temporal relations inside the rules. III) Since our framework is built on temporal relations of intervals, it performs well on interval-based tKGs. Besides, note that timestamp can be considered as a special kind of interval with equal start time and end time. Thus, our definition of temporal relation $TR$ can be also used to describe TR between timestamps, which means our framework also works on timestamp-based tKGs.
>
> -**Writing improvements**
>
> In the introduction of the revision, we analyze the problems of statistical methods including: 1) ignoring the interactions between different rules from the same positive example; 2) unable to deal with the similarity between different rules; 3) their performance on interval-based tKGs is not demonstrated. Then we mention that all these problems are solved by our neural-network-based framework to show that our model is better than other TRL methods.
>
> -**Efficiency study**
>
> We add the efficiency study in Appendix C of the revision. Please check this part in the revision.
>
> -**Reproducibility**
>
> We have submitted the source code as supplementary materials. Please download the code for the confirmation of reproducibility.

---

### Official Review · Reviewer_uk95 · 2022-10-24

**Confidence:** 3
**Correctness:** 4
**Technical Novelty And Significance:** 3
**Empirical Novelty And Significance:** 3
**Recommendation:** 8

**Clarity, Quality, Novelty And Reproducibility:**

Paper is generally well-written and understandable. Experiments are thorough, though additional data or code would be necessary to reproduce. Novelty is medium, being a fairly straightforward extension of existing work.

Seems to build a lot on the Timeplex framework, it’s good that the paper notes the specific differences. It might improve the presentation to present TimePlex and then present the additions, but this is probably not necessary.

Typos / Nits:
Top of page 2, “Wididata12k”

For things like equation 4, a macro like \text{before} should be used instead of writing “before” in a math environment, as the latter looks like a multiplication of 6 variables.

**Strength And Weaknesses:**

Thorough set of well-explained temporal features.

The addition of each feature is well-motivated, though the final set of features is quite complex and might benefit from more concrete running examples through the exposition.

Thorough set of experiments, though rather mixed results. The examples of learned rules are appreciated. Ablation with the “hard settings” in section 6.3 is also appreciated.

**Summary Of The Paper:**

The paper provides a differentiable method for learning temporal knowledge graphs in KGs. While differentiable rule-learning in the context of KG’s already exists, and rule-learning in the context of temporal KG’s already exists, this paper is the first to provide a differentiable rule learning method for temporal KGs.

**Summary Of The Review:**

This is a well executed paper that straightforwardly follows up an existing method (TimePlex) with well motivated additions. The results are mixed but section 6.3 does a good job exploring the method in various settings. This paper would be a good addition to the literature.

---

> ### Author Response · Authors · 2022-11-17
> **Response to Reviewer uk95**
>
> We appreciate the reviewer's comments, which help us improve our paper. In the following we answer the raised issues.
>
> -**Final set of temporal features**
>
> To make the computation of these temporal features more clear, we add details of the temporal feature modelling module in Appendix A of the revision. We also give some concrete examples to illustrate the learning process. Please check this part in the revision.
>
> -**Typos**
>
> We correct all these typos in the revision. Please check the corresponding parts in the revision paper.
>
> -**Reproducibility**
>
> We have submitted the source code and data as supplementary materials. Please download them for the confirmation of reproducibility.

---

> > ### Comment · Reviewer_uk95 · 2022-12-09
> > **A couple notes on appendix A**
> >
> > Thanks for making the additions. Two quick notes on the distributions described in Appendix A:
> >
> > 1) "Duration" is described as a Gaussian random variable. Since Gaussians can be negative, it might be good to clarify how you handle this situation (your formula for intervals could lead to invalid intervals).
> >
> > 2) I believe instead of "Binomial distribution" you mean to say Bernoulli distribution.
> >
> > One small nit: s/Modelling/Modeling/

---

> > > ### Author Response · Authors · 2022-12-09
> > > **Response to Reviewer uk95**
> > >
> > > Many thanks for the notes. Some quick responses are as follows:
> > > 1. For "Duration", we should describe it as a clipped Gaussian random variable, since we use 0 if it is negative.
> > > 2. It should be Bernoulli distribution.
> > > 3. It should be "Modelling".
> > >
> > > We will correct these mistakes in the revision.

---

### Decision · Program_Chairs · 2023-01-20

**Decision:**

Accept: poster

**Justification For Why Not Higher Score:**

While the updated version is clearly better than the previous version, I think the original submission cannot be ignored when evaluating the recommendation type of the paper.

**Justification For Why Not Lower Score:**

Given the characteristics of ICLR, I think we should encourage more submissions like this which might not be very novel but tackles an important problem and makes a significant improvement over an existing work.

**Metareview: Summary, Strengths And Weaknesses:**

The paper received a mixed range of reviews. It seems the paper indeed had some unclear parts as some of the reviewers have pointed out. However, I think many of these were addressed via rebuttal. The problem that the paper is tackling is an increasingly interesting problem in machine learning, and the paper brings a more robust insight into differentiable approaches for learning logic regarding temporal aspect. I think the paper is a great addition to the conference despite its weaknesses.

Strengths:
- Most reviewers agree that the paper's method is well-motivated and new (especially in that previous work has not tackled time range related rules).
- Most reviewers agree that the experiments are reasonable to support the paper's claim.

Weaknesses:
- One of the reviewers was concerned that the paper does not discuss the efficiency of the method.
- Many reviewers were concerned with some unclear parts in the paper, many of which the authors seem to have clarified though.
- Some reviewers were concerned that the proposed method is not sufficiently better than its baseline, TimePlex.

**Note From Pc:**

if the above contains the word "oral" or "spotlight" please see: "oral" presentation means -> notable-top-5% and "spotlight" means -> notable-top-25%. As stated in our emails, we are disassociating presentation type from AC recommendations

**Summary Of Ac-Reviewer Meeting:**

Unfortunately, some of the reviewers who gave low scores did not show up in the virtual meeting. Their concerns were also mostly on the clarity of the paper, most of which seem to have been addressed in the rebuttal with a polished version of the paper. One of those who came to the virtual meeting was eager to see the paper in the conference, and the others did not disagree with the strong supporter.